# Treatment of Vascular Parkinsonism: A Systematic Review

**DOI:** 10.3390/brainsci13030489

**Published:** 2023-03-14

**Authors:** Cristina del Toro-Pérez, Eva Guevara-Sánchez, Patricia Martínez-Sánchez

**Affiliations:** 1Stroke Centre, Department of Neurology, Torrecárdenas University Hospital, University of Almería, 04009 Almería, Spain; 2Faculty of Health Sciences, CEINSA (Center of Health Research), University of Almería, 04120 Almería, Spain

**Keywords:** vascular parkinsonism, treatment, therapy, systematic review, levodopa, vitamin D, repetitive transcranial magnetic stimulation, intracerebral transcatheter laser photobiomodulation

## Abstract

Background and aims: Although the distinction between vascular parkinsonism (VP) and idiopathic Parkinson’s disease (IPD) is widely described, it is not uncommon to find parkinsonisms with overlapping clinical and neuroimaging features even in response to levodopa treatment. In addition, several treatments have been described as possible adjuvants in VP. This study aims to update and analyze the different treatments and their efficacy in VP. Methods: A literature search was performed in PubMed, Scopus and Web of Science for studies published in the last 15 years until April 2022. A systematic review was performed. No meta-analysis was performed as no new studies on response to levodopa in VP were found since the last systematic review and meta-analysis in 2017, and insufficient studies on other treatments were located to conduct it in another treatment subgroup. Results: Databases and other sources yielded 59 publications after eliminating duplicates, and a total of 12 original studies were finally included in the systematic review. The treatments evaluated included levodopa, vitamin D, repetitive transcranial magnetic stimulation (rTMS) and intracerebral transcatheter laser photobiomodulation therapy (PBMT). The response to levodopa was lower in patients with VP with respect to IPD. Despite this, there has been described a subgroup of patients with good response, it being possible to identify them by means of neuroimaging techniques and the olfactory identification test. Other therapies showed encouraging results in studies with some risk of bias. Conclusions: The response of VP to different therapeutic strategies is modest. However, there is evidence that a subgroup of patients can be identified as more responsive to L-dopa based on clinical and neuroimaging criteria. This subgroup should be treated with L-dopa at appropriate doses. New therapies such as vitamin D, rTMS and PBMT warrant further studies to demonstrate their efficacy.

## 1. Introduction

The term vascular parkinsonism (VP) is one of the most controversial in neurology since its introduction in the early 20th century given the heterogeneity of the clinical picture that defines it, the topography of the ischemic lesions that cause it and the response to treatment among patients [1]. This parkinsonism is accompanied by ischemic brain lesions of different characteristics demonstrated by neuroimaging, without findings suggestions of other causes of parkinsonism. Winikates and Jankovic first proposed clinical criteria for vascular parkinsonism (VP) in 1999 [2]. New, stricter criteria based on a clinicopathological study were defined in 2004 [3], although a definitive diagnosis can only be reached by autopsy [4].

Currently, VP encompasses a heterogeneous set of clinical pictures in which the predominant syndrome is similar to parkinsonism but without meeting the necessary diagnostic criteria, which can present in various forms. VP has also been termed “lower body parkinsonism” because it can manifest as predominant parkinsonism of the lower extremities, with difficulty walking, absence of tremors and minimal or no response to levodopa treatment, especially in hypertensive patients. However, cases with clinical features difficult to distinguish from idiopathic Parkinson’s disease (IPD) have also been described, with a response to levodopa, even without evidence of Lewy bodies in post-mortem studies [4].

Classically, it has been considered that VP did not show a good response to levodopa treatment. However, a study published in 2004 showed that a subgroup of patients with vascular lesions in or near the nigrostriatal pathway could be responders to levodopa regardless of VP characteristics [5]. Following this, several studies have tried to identify clinical or radiological features that might explain or anticipate a good treatment efficacy, with different response rates described, and therapies other than levodopa or dopaminergic agonists have also been tested. Given the limitations of VP treatment and the emergence of new therapeutic strategies since the last meta-analysis [6], the present systematic review has been performed.

## 2. Materials and Methods

### 2.1. Search Strategy

This paper follows the guidelines according to the preferred reporting items for systematic reviews and meta-analysis protocol (PRISMA-P) [7]. It was registered in the PROSPERO international database of prospectively registered systematic reviews (CRD 42021250195). Pubmed, Scopus and Web of Science electronic databases were searched for articles in English or Spanish, published in the last 15 years until April 2022 and with the following criteria: randomized clinical trials, cross-sectional, case-control and cohort observational studies including patients with VP and treatment of VP, analyzing differences between given therapies and their efficacy. Case reports and animal-model studies were excluded. The search query was: (“Parkinson Disease, Secondary” OR “Parkinsonism, Symptomatic” OR “Symptomatic Parkinson Disease” OR “Symptomatic Parkinsonism” OR “Secondary Parkinsonism” OR “Parkinson Disease, Symptomatic” OR “Parkinsonism, Secondary” OR “Parkinson Disease, Secondary Vascular” OR “Secondary Vascular Parkinson Disease” OR “Atherosclerotic Parkinsonism” OR “Parkinsonism, Atherosclerotic” OR “Parkinson’s disease” OR “PD” OR “Lower Body Parkinsonism” OR “Pseudo-parkinsonism” OR “acute parkinsonism” or “vascular parkinsonism”) AND (“vascular” OR “stroke” OR “brain ischemia”) AND (“treatment” OR “disease management” OR “Therapeutic” OR “Therapy” OR “Therapies” OR “Treatments”). In addition to the database search, a manual revision of the reference lists of all relevant articles was performed to identify additional studies of interest.

### 2.2. Selection of Studies

Two researchers (CT and EG) separately reviewed the titles and abstracts of the retrieved articles to determine the presence of the abovementioned criteria. Disagreements were solved by the consensus of a third author (PM). Two investigators (CT and EG) separately reviewed the titles and abstracts of the retrieved articles to determine the presence of the abovementioned criteria. Disagreements were resolved by consensus of a third author (PM). These results were transferred to Rayyan (https://www.rayyan.ai/), accessed on 25 April 2022.

For systematic, independent screening for exclusion or inclusion by two reviewers (CT and EG). Duplicate entries, studies on diseases other than vascular parkinsonism or studies evaluating another aspect of vascular parkinsonism other than its treatment, papers not written in English or Spanish, publications that were not research studies, and any other articles that did not fit the scope of the review were excluded.

### 2.3. Data Extraction

After manuscript selection, the following information was extracted: the number of participants and socio-demographic characteristics, the assessed scales and the evaluation protocol or diagnostic strategies, the type of vascular lesion, response to evaluated treatment and the major findings reported. We expected to find a limited number of studies that could eventually be included in the review.

### 2.4. Quality Assessment

To improve the quality of detection of the risk of bias in non-randomized studies, these will be assessed using the Newcastle–Ottawa scale, with a subsequent comparison with the STROBE scale used in the last systematic review of 2017. The Cochrane risk-of-bias tool for randomized trials (RoB 2) will be used for randomized studies [6,8,9,10].

## 3. Results

The Databases search yielded 4738 results. Overall, 4687 publications involving different pathologies were excluded. After removing duplicates, 59 publications were screened for eligibility. Of them, 8 studies were identified through the references of the principal records. A total of 46 studies were excluded for the following reasons: publications that evaluated different pathologies, no evaluation of response to treatment, systematic reviews, studies in languages other than Spanish or English, experimental studies with animals or studies prior to the last 15 years.

A PRISMA flow diagram is shown in Figure 1. After reading the articles and removing duplicates, a total of 12 original studies were finally included in the systematic review and are summarized in Table 1 and Table 2. Table 3 shows the studies performed using neuroimaging studies. Finally, a meta-analysis was not performed as we have not found studies on the response to levodopa in VP since the last systematic review and meta-analysis of 2017 and there are insufficient studies on other treatments to perform it on another treatment subgroup.

As no randomized studies were found in the search, the Cochrane Collaboration tool for assessing risk of bias was not used. The risk of bias in the included studies was assessed using the Newcastle–Ottawa Scale. This can be seen in Figure 2 together with a comparison of the results of this scale with the STROBE scale used in the previous systematic review.

## 4. Treatments

### 4.1. Levodopa

A cross-sectional study assessed the characteristics and response to levodopa in 17 patients with a diagnosis of VP [14]. VP was divided into four types based on Fenelon and Houéto classification [23]: (1) VP identical to IPD, (2) Unilateral after a contralateral vascular lesion, (3) Atypical parkinsonism and (4) Parkinsonian gait disorder. They added three categories depending on the course: (1) Rapidly progressive (worse before a year from its onset), (2) stable and (3) slowly progressive (worsening after a year from its onset). Response to levodopa was based on the percentage of reduction on Part III of MDS-UPDRS and the Hoehn and Yahr stage (HY), evaluated in an “off” state (12 h after interruption of levodopa) and an “on” state (1 h after levodopa). The patients had been treated with levodopa at a mean dose of 530.9 ± 218.2 mg/day for a mean period of 2.9 years. There was a mean of 5.8 ± 4.4 point reduction in UPDRS Part III after levodopa, with no change in the HY stage. Most patients had a poor response to the drug and no complications of levodopa were seen, such as dyskinesia or fluctuation. Two years later, the same first author designed a case-control study to further study VP, comparing baseline, imaging and response characteristics to levodopa compared to IPD patients. He observed that 33.3% of IPD patients with freezing of gait “off” (50%) responded to levodopa, whereas no patients with VP responded. The percentage of patients responding to levodopa was lower in the VP group, although the mean MDS-UPDRS part III score did decrease [19].

Two studies have relied on gait to assess VP response to levodopa. The first of them [24] is a non-blinded, non-randomized, case-control study adding levodopa response to increase the accuracy of the differential diagnosis between IPD and VP according to gait characteristics and response to treatment, based on a previous study that succeeded in discriminating between IPD, VP and healthy controls by gait assessment thanks to machine learning strategies (accuracy for distinguish IPD and VP was 50–63.3%) [22,24,25]. 14 VP and 15 IPD were included, excluding patients with resting tremors, dementia CDR > 2, musculoskeletal disease and an HY stage. Similar to previous studies, 36 controls were added for the normalization process of gait data. Patients were evaluated after 12 h in the “off” state and after 60 min after taking >50% over their usual dose of levodopa (“on” state). Speed, stride length and foot clearance were the independent variables included to predict differences between patients with and without IPD, based on previous studies [26,27,28]. The results showed increased discrimination due to levodopa comparing “on” and “off” status, achieving IPD diagnostic accuracies of 86% ± 7.12, the sensitivity of 80% ± 16.33 and 90% ± 20, as well as a VP diagnostic accuracy of 72.8% without levodopa testing. These results show that the inferior response to levodopa treatment in VP is also reflected in gait. In the second one, Gago M.F. et al. evaluated the effect of levodopa on postural stability [17]. Two groups (VP and IPD) with normal retropulsion tests were included. The included IPD patients were of the akinetic-rigid type. Both groups were age-matched since gait is altered by age. Wearable sensor-based gait was compared when patients were in their best “on” state with gait “off”. The best dopaminergic therapy to reach their best “on” state was assessed over the three months prior to the start of the study. Five VP and 10 PD were included. The IPD group had better MoCA scores, gait, lower UPDRS III scores, easier getting out of a chair and global spontaneity of movement after levodopa treatment, with a motor benefit of 19% of VP patients vs. 57.5% of IPD patients.

On the other hand, several studies have added imaging studies for the study of the response to levodopa in VP (Table 3). In the study by J. Navarro-Otano. et al. the aim was to add diagnostic accuracy to the difference between VP and IPD using the University of Pennsylvania smell identification test (UPSIT) to cardiac imaging by 123I-MIBG [18]. Patients were diagnosed with VP using the criteria of Ziljmans 2004, and patients with IPD using the criteria based on Huges, 1992. The discrimination ability between IPD and VP of the tests as well as the response to treatment were studied. A greater response to levodopa was observed in patients with IPD compared to VP (100% vs. 14.3%). However, the response rates to levodopa were different within the group of VP patients according to the results of the tests performed. No VP patient with a normal H/M ratio in the 123I-MIBG test (not suggestive of IPD) responded to levodopa, whereas 28.6% of VP with a low ratio presented a good response (*p* = 0.0462). Of note, 123 I-MIBG SPECT can be positive in diabetic patients.

It has been seen that VP patients with nigrostriatal dysfunction assessed by PET study showed significantly better response to levodopa, although VP patients with vascular lesions in the basal ganglia were excluded [20]. Greater than or equal to 30% improvement in UPDRS motor score was observed in 40% of patients with pathological PET vs. 4.5% of patients with normal PET. Also, a partial response (improvement between 10–30% UPDRS motor scale improvement) was observed in 20% vs. 13.6%. Finally, poor response to levodopa was observed in 40% of patients with pathologic PET versus 81.8% with normal PET (change of less than 10%. *p* = 0.036). Clinical differences between patients in whom nigrostriatal dysfunction was observed and between those without this dysfunction did not predict response to levodopa. MRI imaging also failed to predict response to levodopa, with no differences in the degree and regional distribution of white matter lesions between responders and non-responders.

On the contrary, in the study by Benítez-Rivero et al. when clinical characteristics and levodopa response were analyzed with the results of normal or pathological 123 I-FP-CIT SPECT in patients with VP (pathological in 67.5% vs. 100% of patients with IPD), pathological SPECT was only associated with the presence of falls and not with levodopa response [15]. In patients with VP, 47.9% of patients who received levodopa treatment had an improvement vs. 100% of patients with PD.

Zijlmans J. et al. performed a case-control study that aimed to compare by [123I] FP-SPECT uptake: (1) pre-synaptic dopaminergic function VP vs. EPI; (2) acute-onset VP vs. insidious-onset VP; (3) severity of parkinsonism and (4) response to levodopa [11]. It included 13 VP (6 with acute onset, 7 with progressive onset), 14 controls and 14 IPD. It included 13 VP (6 with acute onset, 7 with progressive onset), 14 controls and 14 IPD. Withdrawal of dopaminergic therapy was performed 12 h before the levodopa challenge test. There was a good response in one patient, transient in two, poor in three and uncertain in five. No patients with VP had an excellent response to levodopa, with no difference between acute-onset VP and insidious-onset VP. 123I] FP-SPECT uptake does not correlate with response to levodopa based on UPDRS III scale reduction.

Finally, in the Antonini et al. study [12], a greater negative response to levodopa was observed in VP concerning IPD. A total of 47.8% of VP patients responded to treatment, being their negative response associated with symmetrical symptom onset (*p* < 0.001), HY status (negative 2.43 ± 0.8 vs. positive 2.16 ± 7; *p* = 0.007), absence of dyskinesia (*p* = 0.04) and hypertension and diabetes (*p* = 0.04 and *p* = 0.04). Higher HY status was associated with hypertension and smoking (*p* = 0.005; *p* = 0.05). The strongest predictor variables for a negative response to levodopa (failure to achieve > 30% improvement on the UPDRSIII scale with levodopa 500 mg/day for more than 3 months) were hypertension (systolic blood pressure > 140 mmHg and/or diastolic blood pressure > 90 mmHg) (*p* = 0.022), basal ganglia lesions (*p* = 0.045) and normal FP-CIT SPECT uptake (*p* < 0.001). More of these and other vascular risk factors (family history, hyperlipidemia, heart disease and hypotension) predicted a negative response to chronic levodopa. In patients with pathological uptake on FP-CIT SPECT, vascular lesions in the basal ganglia predicted a negative response to levodopa, and hypertension and vascular lesions in infratentorial areas were associated with worsening disease (*p* = 0.007; *p* = 0.045). VP patients with normal FP-CIT SPECT showed no effect with levodopa. However, in IPD patients with normal FP-CIT SPECT, although they had a worse response to levodopa than those with pathological uptake (48% vs. 93%, *p* < 0.001), we did find a percentage with response to levodopa that was not found in those VP with normal FP-CIT SPECT. Cerebral vascular disease is found to be associated with increased severity of parkinsonism and poor response to levodopa, especially in patients with non-pathological FP-CIT SPECT.

### 4.2. Vitamin D

Sato et al. designed a 2-year case-control study [13]. The objective was to reduce falls in patients with VP and IPD by vitamin D supplementation of 1200 IU ergocalciferol in vitamin D-deficient patients (mean vitamin D at baseline 22 nmol/L, low compared to the reference range of the normal Japanese population). It is speculated that the protective effect of vitamin D is due not only to its benefits on bone mineral density but also to the enhancement of atrophy of type II muscle fibers, which prevents falls [29,30]. In addition, one study showed that deterioration of muscle function can be observed before signs of bone density loss [31].

Between 92 IPD and 94 VP patients participated. No changes in diet, physical activity or medication that could alter bone or calcium were introduced. Sunlight exposure and muscle strength were assessed and fall schedules and medication adherence were recorded. No differences in baseline clinical characteristics were found. After 12 and 24 months, no differences were found in PD patients, while the percentage of falls was reduced from 34% to 16% in the VP group (*p* < 0.001). A significant increase in muscle strength was observed in both groups. This study adds evidence to the fact that falls have a different etiology in VP and PD, with a possibly greater role of muscle weakness in VP than in PD.

### 4.3. Repetitive Transcranial Magnetic Stimulation (rTMS)

rTMS has shown beneficial outcomes in bradykinesia and UPDRS scales in IPD [16]. The study of Jang et al. aims to improve gait in VP, based on the mean timing measured in seconds of 10 metres walk and the improvement of UPDRS scores. This study was unblinded and non-randomised. The leg region was identified for each patient by motor-evoked potential. Five patients were included, with 4/5 presenting a headache response to simple analgesics as adverse events. Improvement was observed after 4 weeks of treatment that did not persist in week 6. UPDRS score reduction was observed at weeks number 2, 4 and 6 after rTMS. Also, two 7-point scales were performed based on the Patient’s Global Impression of Change and Clinicians’ Global Impression of Change, with a significant increase in both of them after rTMS.

### 4.4. Intracerebral Transcatheter Lase Photobiomodulation Therapy (PBMT)

Cerebral small vessel disease progresses causing leukoaraiosis. Cerebral hypoperfusion and hypoxia stimulate angiogenesis with the development of collateral capillary supplementation [32], facilitating angiogenesis neurogenesis [33]. Intracerebral transcatheter PBMT has shown good results in the treatment of stroke, neurodegenerative diseases, trauma and depression [34,35,36].

The study by Maksimovich et al. aims to evaluate intracerebral transcatheter PBMT as a treatment for Binswanger’s disease and VP, using a case-control study [21]. Sixty-two subjects with VP and 27 with BD were enrolled. After PBMT the VP patients continued dopaminergic therapy (levodopa 250 mg three-four times daily + Amantadinum 100–200 mg daily). The control group of the VP arm was prescribed the same dopaminergic therapy. In the first 6 months after therapy, 94.6% of the VP case group vs. 56% of the controls had significant improvement in mental and motor functions. 100% of cases vs. 52% of controls in the VP group had improvement in blood flow measured by scintigraphy (SG) and rheoencephalography (REG) as well as narrowing of the subarachnoid space assessed by CT and MRI vs. 0% of controls. After 8 years, the restoration of mental and motor functions remained at the same percentage in the case group, while the patients who improved in the control group suffered a clinical worsening at 12–24 months. Improvement in OS and REG was maintained in 94.6% of patients versus 52% in the control group. CT and MRI showed a decrease in involutional changes in 91.89% and a narrowing of the sylvian fissure in 86.5%, while 100% of the control group had greater involutional changes.

## 5. Discussion

Classically, VP has been considered a homogeneous entity with poor response to levodopa treatment. However, the reviewed studies suggest that VP is a heterogeneous entity that should be properly subclassified to identify those patients with a response to levodopa. Several treatments have been added in recent years as possible adjuvants and even as effective treatments, but further studies are needed to confirm their efficacy.

Levodopa resistance has been considered a useful feature to distinguish between PD and VP. However, despite not showing excellent response to levodopa in a high percentage of patients, a decrease in part III of the UPDRS scale has been observed in this review. Moreover, some patients with VP have been shown to have clinical benefits from levodopa treatment for several months [5,37], and even an excellent positive response to levodopa has been described in pathologically confirmed VP [5]. Furthermore, the fact that the clinic cannot reliably distinguish patients with nigrostriatal dopaminergic denervation (NDD) a Lee et al. study provides additional evidence that in case of non-response to levodopa in patients with VP, levodopa should be increased to the maximum tolerable dose (up to 1 g L-dopa daily for 3 months) [11].

S Benítez-Rivero, et al. together with Ziljmans et al. [15,38], reported dopaminergic deficits in patients with VP, sometimes as marked as in patients with IPD. Other studies did not find this dopaminergic deficit [39]. Also, neuropathology studies have shown a heterogeneous clinical presentation of VP, sometimes with an overlap between VP and IPD that increases in VP patients with a response to levodopa [3,5]. Therefore, although the distinction between both entities by clinical features is widely described, it is not uncommon to find parkinsonisms with overlapping clinical and neuroimaging features and even in the response to levodopa treatment. Although structural imaging based on magnetic resonance imaging (MRI) or computed tomography (CT) shows vascular lesions in all VP, these are also prevalent in IPD patients in up to 25%, their contribution to the clinical features is unknown [12,15]. Levodopa has shown very variable response rates in VP patients in different studies, but almost always much lower than the response rates in patients with IPD. It has been described that a presynaptic dopaminergic deficit evidenced by SPECT and corresponding to ischemic lesions in MRI, simulating the pathological mechanism of IPD, could have a response to levodopa administration. That is why several studies try to delve into their clinical and imaging features to facilitate the differential diagnosis and especially to identify those patients who may benefit from treatment.

In a cross-sectional study of 15 patients with VP, 15 patients with EPD and 9 healthy subjects, the usefulness of olfactory function assessment measured with the University of Pennsylvania Smell Identification Test (UPSIT), cardiac SPECT with 123 I-meta-iodobenzylguanidine (123 I-MIBG) and SPECT with I-FP-CIT assessed by a blinded nuclear medicine specialist was studied [18]. The heart-to-mediastinum ratio was higher in VP versus IPD, with discrimination between VP and IPD under the ROC curve of 0.85. UPSIT scores were similar between VP and IPD. However, patients with normal H/M radius were more likely to have higher UPSIT scores. No VP H/M normal ratio patients (non-suggesting IPD) responded to levodopa, whereas 28.6% of VP with a low ratio presented a good response with statistical significance. As previously mentioned, it is worth noticing that 123 I-MIBG SPECT can be positive in diabetic patients. Other studies did find higher UPSIT scores in patients with VP vs. IPD [40], but the response to levodopa as a function of UPSIT scores was not studied.

Other studies show that a higher burden of cerebral vascular disease is associated with more severe parkinsonism and a negative effect of levodopa, especially in patients with non-pathological FP-CIT SPECT [12]. The location of vascular lesions has also been shown to be related to different clinical features of patients with VP and their response to treatment; Antonini et al. showed that the lesion most strongly predicting a negative effect of levodopa is in the basal ganglia [12], and Benítez-Rivero found that territorial infarction was related to lower response to treatment [15]. The study of Benítez-Rivero et al. found no association between pathological SPECT imaging in VP patients and their response to levodopa and no association between CT/MRI and SPECT findings [15]. On the contrary, the VADO study found several differences in terms of structural imaging with CT or MRI according to the SPECT result [12]. IPD patients with normal FP-CIT SPECT had a worse response to levodopa, a higher HY scale score and greater periventricular leukoaraiosis, while pathological FP-CIT SPECT was associated with vascular lesions in basal ganglia and infratentorial regions. Classic VP clinic (symmetrical onset, higher disease severity based on HY stage, negative response to levodopa) was associated with higher vascular scores [4,12,41]. Interestingly, despite vascular burden, IPD patients with abnormal MRI and pathological SPECT FP-CIT showed a good response to levodopa. These findings are consistent with a worse response to levodopa in patients with non-classical IPD clinic, as well as opening the possibility that those IPD patients with higher vascular lesion burden get a worse response given the irruption of striatal pathways [12]. Other studies have also added evidence that abnormal uptake on FP-SPECT [123I] correlates with disease duration and severity of parkinsonism [4,41]. Nevertheless, some studies show that a chronic response to levodopa can be seen in 50% even in those patients with a normal SPECT FP-CIT [11,12,15,18]. A negative response to levodopa was associated with the symmetrical onset of symptoms characteristic of VP, as well as an absence of dyskinesia (and thus the response to levodopa), hypertension and diabetes [12]. 

The [123I] FP-SPECT study performed by Zijlmans J. et al. [11] showed a lower uptake in both acute-onset and progressive-onset VP patients versus controls, as well as a higher caudate/putamen ratio. However, interhemispheric asymmetry did not differ between VP and controls nor between both VP groups. This is further evidence alongside the study by Lee et al. [20] that VP patients have a significant presynaptic dopaminergic deficit. Postmortem studies in which nigral cell loss and substantia nigra gliosis in VP occur in a similar pattern to IPD support these findings, with greater involvement of the rostral parts of the striatum compared to the lateral striatum [20,41]. In this study [123I] FP-SPECT uptake did not correlate with response to levodopa based on UPDRS III scale reduction. Lee et al. suggested that leukoaraiosis in VP may cause NDD detectable by [(18)F] FP-CIT PET. Clinical differences between VP NND+ and NND− did not predict levodopa response but the presence of NDD did predict a better response to levodopa treatment [20]. These findings are consistent with those of the study by Antonini et al. in which patients with VP with abnormal MRI and normal FP-CIT SPECT had a poor response to levodopa. In this study, 90% of patients (including IPD and VP) with normal FP-CIT SPECT showed no effect with levodopa [12].

The research designed by Fernandes et al. also shows lower response to levodopa treatment in the VP group in terms of gait disorders. It also adds a useful tool for the differential diagnosis between both entities through the effect of treatment on various gait characteristics assessed by machine learning [22]. The study by Gago M.F. et al. also showed the validity of the gait study of patients with IPD and VP to differentiate both entities, especially in the “on” state. It also evidenced the better response to levodopa treatment in terms of gait disturbances in patients with IPD versus VP. However, it should be noted that some patients with VP did benefit in this respect with treatment, albeit to a lesser extent [22].

Regarding vitamin D treatment, Sato et al. showed a significant difference in the bivariate analysis between VP and IPD in the number of falls per subject over the 2 years after treatment with 1200 IU of ergocalciferol per day, with an increase in muscle strength in the lower extremities that was also observed in both groups. Therefore, this study suggests that vitamin D decreases falls and hip fractures in VP by increasing muscle strength and should be confirmed with further studies that include an analysis adjusted for confounding variables.

Treatment by repetitive transcranial magnetic stimulation (rTMS) at 5 Hz on 5 consecutive days showed improvement in a timed 10-m walk (T10MW), motor portion of the Unified PD Rating Scale (UPDRS-III), global impression of medical change (CGIC), and global impression of patient change (PGIC), up to 6 weeks after rTMS. The treatment was well tolerated, and all patients completed the study. This work demonstrated for the first time that 5 sessions of rTMS could measurably improve gait for up to 6 weeks without significant side effects, so it could be a potentially useful adjunct in the rehabilitation of VP patients and warrants further investigation as these results need to be validated with other studies with a control group and multivariate analysis.

More recently, treatment with intracerebral transcatheter laser photobiomodulation therapy (PBMT) has been successfully studied for VP. After 8 years the restoration of mental and motor functions was maintained with the same percentage in the testing group whereas the control group suffered a clinical worsening. Improvement in blood flow persisted in virtually all patients with VP, twice as many as in control patients. Likewise, a decrease in the signs of brain involution was observed, while 100% of the control group presented greater involutionary changes during the observation period. Despite obtaining encouraging results, this study does not specify the definition of VP and lacks control of the treatment effect through a blinded study and a confounder-adjusted analysis.

It is important to highlight that the articles included in this systematic review show a high risk of bias according to the Newcastle–Ottawa scale. This bias has been compared with that described in the previous systematic review which used the STROBE checklist with a lower bias rate. Despite this, most of the studies also showed a high risk of bias even when using this other scale. This bias clearly increases when performed in response to levodopa, which was not the primary endpoint in several of the articles. Few articles make a good case-control comparison using a statistical study adjusted for confounding variables probably due to the low number of patients in some of them and some articles have no control group, as can be seen in the comparability part of Table 1. Neither have they been performed in a blind manner for the patient or physician providing the medication. Only one of them did not mention the diagnostic criterion of VP. Both systematic reviews show that there is a lack of high-quality evidence regarding the treatment of VP.

## 6. Conclusions

The response of VP to different therapeutic strategies is modest. However, there is evidence that a subgroup of patients can be identified as more responsive to L-dopa based on clinical and neuroimaging criteria. This subgroup should be treated with L-dopa at appropriate doses. New therapies such as vitamin D, rTMS and PBMT deserve further studies to demonstrate their efficacy.

## Figures and Tables

**Figure 1 brainsci-13-00489-f001:**
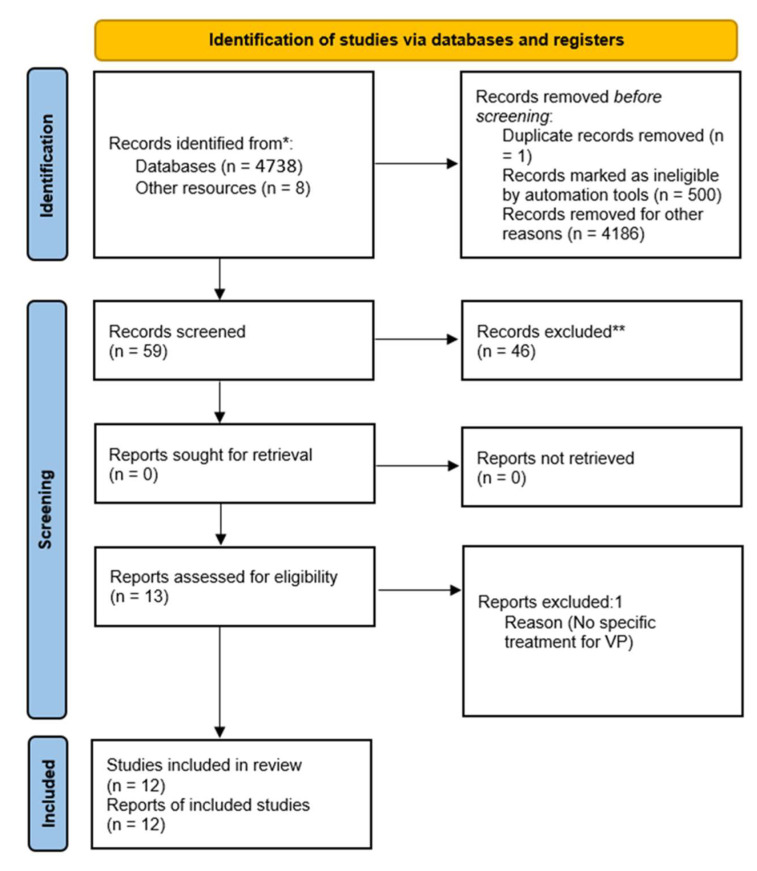
PRISMA Flow Diagram. * Pubmed, Scopus and Web of Science electronic databases. ** Studies excluded for the following reasons: publications that evaluated different pathologies, no evaluation of response to treatment, systematic reviews, studies in languages other than Spanish or English, experimental studies with animals [11,12,13,14,15,16,17,18,19,20,21,22].

**Figure 2 brainsci-13-00489-f002:**
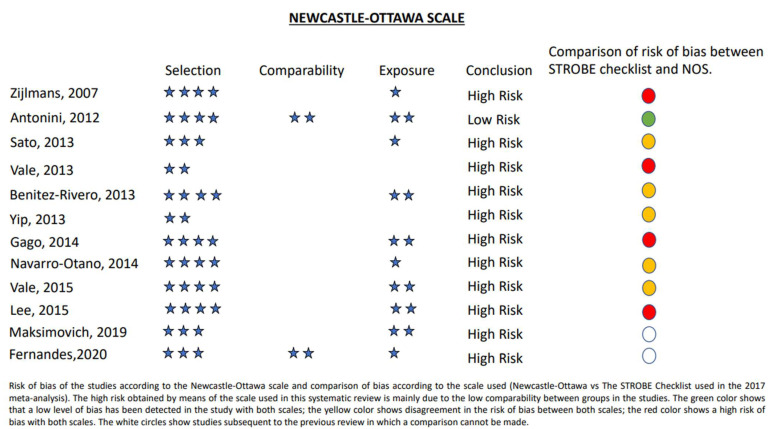
Newcastle–Ottawa scale and comparison of bias with STROBE scale.

**Table 1 brainsci-13-00489-t001:** Studies methodology and clinical-demographic characteristics.

Study	Design	Mean Age (n)	Main Symptoms	Comparative Group (vs. VP)	Follow-Up	Blinded	Diagnostic Criteria for VP
**Zijlmans, J. et al., 2007** [11]	Case-control	Group I (VP): 74.1 years ± 11.5 (13) Group II (IPD): 66.0 ± 14.5 (14) Group III (controls): 66.3 ± 18 (14)	Gait disorder, acute contralateral bradykinetic rigid syndrome, cognitive dysfunction	VP/IPD	No	Nuclear Medicine specialists	Zijlmans et al.
**Antonini, A. et al., 2012** [12]	Case-control	Group I (SPECT no pathological): 72.8 ± 4.8 (59–80) (28) Group II (SPECT pathological): 72.6 ± 5.7 (48)	Lower body parkinsonism	VP/IPD	No	Radiologists and Nuclear Medicine specialists	Zijlmans et al.
**Sato, Y., et al. 2013** [13]	Case-control	Group I (VP): 73.9 ± 6.2 (94) Group II (IPD): 73.6 ± 5.9 (92)	Bradykinetic rigid syndrome, rest tremor	VP/IPD	2 years	A therapist that evaluated muscle strength	Zijlmans et al.
**Vale, T.C. et al., 2013** [14]	Case series	Group I (VP): 75.8 ± 10.1 (17)	Lower body parkinsonism, pyramidal signs, urinary incontinence	Their selves	No	No	Zijlmans et al.
**Benítez-Rivero, S. et al., 2013** [15]	Case series to correlate image to VP clinic. Case control to find clinical and image differences between PD and VP	Group I (VP): 72.6 ± 6.8 (106) Group II (IPD): 55.3 ± 12.6 (280)	Gait disorder, postural tremor, mixed tremor, rest tremor, falls, postural instability, dysphagia, urinary incontinence, cognitive dysfunction, emotional lability	VP/IPD	5 years	Nuclear Medicine specialists	Zijlmans et al.
**Yip, C.W. et al., 2013** [16]	Case series	Group I (VP): 64.2 (5)	Bradykinetic rigid syndrome, tremor, postural instability	No	6 months	No	Winikates et al.
**Gago, M.F. et al., 2014** [17]	Case-control	Group I (VP): 77 (5) Group II (IPD): 73 (10)	Worse MoCA and UPDRS III scores, gait impairment, difficulty getting up from the chair and low global spontaneity of movement	VP/IPD	No	No	Zijlmans et al.
**Navarro-Otano, J. et al., 2014** [18]	Case-control	Group I (VP): 68.11 ± 8.2 (15) Group II (IPD): 66.2 ± 9.5 (15) Group III (controls): 66.2 ± 8.2 (9)	Gait disorder, postural tremor, falls, postural instability	VP/IPD/controls	No	Nuclear medicine specialists	Zijlmans et al.
**Vale, T.C. et al., 2015** [19]	Case-control	Group I: (VP) 75.7 ± 10.4 (15) Group II: (IPD) 67.3 ± 7.5 (30)	Lower body parkinsonism, pyramidal signs, instability urinary incontinence	VP/IPD	No	No	Zijlmans et al.
**Lee, M.J. et al., 2015** [20]	Case-control	Group I: (no pathological) 75.77 ± 6.16 (22) Group II: (pathological) 75.15 ± 6.75. (20)	Gait disorder, postural tremor, resting tremor, falls, postural instability, urinary incontinence supranuclear palsy, dysphagia, emotional lability	NDD+/NDD−	No	No	Zijlmans et al.
**Maksimovich, I.V. et al., 2019** [21]	Case-control	Group I (VP): 52–80 (37) Group II (control group): (25)	Cognitive dysfunction	VP/ Binswanger Disease/ controls	8 years	No	Does not specify
**Fernandes, C. et al., 2021** [22]	Case-control	Group I (VP): 80.53 ± 4.63 (14) Group II (IPD): 76.60 ± 4.29 (15) Group III (controls): 52.76 ± 22.91 (34)	Gait disorder	VP/IPD/controls	No	No	Zijlmans et al.

**Table 2 brainsci-13-00489-t002:** Study methodology (continuation) and main results.

Study	Used Scales Image Testing	Type of Vascular Lesion Specified	Primary Endpoint	VP Treatment as Primary Endpoint	Definition of Treatment Response	Response to Treatment
**Zijlmans, J. et al., 2007** [11]	UPDRS III [123I] FP-CIT SPECT	In or near areas that can increase the basal ganglia motor output or decrease the thalamocortical drive directly (substantia nigra in one, globus pallidum/putamen area in the others). Extensive subcortical white matter lesions	To compare pre-synaptic dopaminergic function VP vs IPD; VPa vs VPi and if severity and response to levodopa can be related to pre-synaptic dopaminergic function	Yes	Based on the mean % reduction in motor UPDRS	(L-dopa) Mean reduction in motor UPDRS in Group I (VP patients): 14% “Good” response: 0.07%
**Antonini, A. et al., 2012** [12]	UPDRS III, UPDRS II, Y&H, DAT SCAN	Periventricular hyperintensities, lesions in hemispheric white matter, basal ganglia, infra-tentorial foci	Clinical and neuroimage profile	No	≥30% changes in total UPDRS motor scores from the baseline	(L-dopa) Negative response: Group I (VP patients): 68.4% Group II (IPD patients): 40%
**Sato, Y. et al., 2013** [13]	Barthel index, SSS arm score, SSS leg score	Cerebral infarction/Cerebral hemorrhage	Clinical profile	Yes	The number of falls per person and incidence of hip fractures	(Vitamin D supplementation) VP patients: 59% reduction in falls IPD patients: 0% reduction in falls Increase of strength in both groups (does not provide details)
**Vale, T.C. et al., 2013** [14]	DSM-UPDRS, HY, MMSE, FAB, EIS, Pfeffer, Katz, NINDS-AIREN	Substance nigra, White matter disease, Multiple lacunar infarcts	Clinic and radiological profile	No	Based on the percentage of reduction in Part III of DSM-UPDRS and Hoehn-Yahr	(L-dopa) Improvement in part III DSM-UPDRS: 5.8 ± 4.4 (Efficacy is based on mean scale score reduction, no control group)
**Benítez-Rivero, S. et al., 2013** [15]	UPDRS, HY, Stchelten’s scale DAT SCAN,	Supratentorial lesions: Subcortical basal gabglia>thalamus>internal capsule Infratentorial	(A) To find clinical and image (SPECT) differences between IPD and VP. (B) Among VP patients, to study possible clinical features related to SPECT or structural image (CT or MRI)	No	Does not specify criteria for responsiveness	(L-Dopa) Group I (VP patients): 47.9% Group II (IPD patients): 100%
**Yip, C.W. et al., 2013** [16]	UPDRS rTMS	Multiple lacunar infarcts, lentiform nuclei, caudate, Multiple subcortical lesions	Gait improvement	Yes	Mean timing measured in seconds of 10 m walk and the improvement of UPDRS score	(rTMS) At 4 weeks post-rTMS:11.9%, At 2 weeks post-rTMS: 6.8%, Not statistically significant by 6 weeks For the UPDRS post-rTMS over time: 11.8%
**Gago, M.F. et al., 2014** [17]	MDS-UPDRS III, MoCA	Subcortical or basal ganglia lesions	Clinical improvement on postural stability	Yes	Percentage of the difference between “off” and “on” states	(L-dopa) Group I (VP patients): 19% Group II (IPD patients): 57.5%
**Navarro-Otano, J. et al., 2014** [18]	UPDRS, HY 123I-MIBG cardiac gammagraphy, UPSIT, DaT-SPECT	Decreased uptake with a pattern typical for IPD (symmetric or asymmetric levodopa uptake reduction or absent uptake) or decreased uptake pattern non-typical of PD (as a local or patchy defect where cerebral MR imaging showed an ischemic lesion)	To ascertain the clinical value of 123I-MIBG cardiac gammagraphy, UPSIT and DaT-SPECT to diagnosis	No	Levodopa response was codified as good, partial and absent, does not specify criteria for responsiveness	(L-dopa) Group I (VP patients with normal H/M ratio): 0% Group II (VP patients with low H/M ratio): 28.6% In total good response: 14.3% Group III (IPD patients): 100%
**Vale, T.C. et al., 2015** [19]	MDS-UPDRS, MMSE, FAB, EXIT25, Hachinski scale, Katz index, Pfeffer, FOG-Q, HY	Extensive white matter disease, Multiple lacunar infarcts	Clinic and radiological profile	No	Based on the percentage of reduction in Part III of MDS-UPDRS	(L-dopa) Yes Not percentages
**Lee, M.J. et al., 2015** [20]	MMSE, UPDRS III [18F] FP-CIT PET, MRI	Moderate or severe white matter lesions in the lobar subcortical or periventricular regions Deep subcortical lesions in frontal, temporal, parietal and occipital regions	Clinical and MRI findings that indicate NDD	No	≥30% changes in total UPDRS motor scores from the baseline	(L-dopa) Group I (Normal uptake): 4.5% Group II (reduced uptake): 40% In total good response: 44,5%
**Maksimovich, I.V. et al., 2019** [21]	Clinical Dementia Rating scale, MMSE, BI MRI, CT, SG, REG, MUGA.	Signs of brain involutional changes, Subarachnoid space expansion, Nonocclusive hydrocephalus signs, Local focal subcortical demyelization, Leukoaraiosis signs	Clinical and image improvement	Yes	Mental and motor functions, an improvement in blood flow measured through SG and REG and a narrowing of the subarachnoid space	(PBMT) Group I (case group VP with PBMT):94.60% Group II (control group VP without PBMT): 56.00% Group III (case group BD with PBMT): 53.85%
**Fernandes, C. et al., 2021** [22]	CDR, Hoehn-Yahr CNNs	Does not specify	Clinical improvement	Yes	Based on gait time series with and without the influence of levodopa medication	(L-dopa) Group I (VP patients): 79.33% Group II (IPD patients): 82.33% Group III (controls): 86%

(123I)-MIBG cardiac gamma-graphy: (123)I-metaiodobenzylguanidine on cardiac gammagraphy; (123I) FP-CIT SPECT: Single Photon Emission Tomography with 123Ioflupane; 18F-FDG-PET: Fluorodeoxyglucose labeled with 18F Positron Emission Tomography; CDR: Clinical Dementia Rating; CNNs: Calibrated Neuropsychological Normative System; CT: Computerized Tomography; DaT: Dopamine Transporter; DSM/MDS: Diagnostic and Statistical Manual of Mental Disorders; EIS: Executive Interview Scale; EXIT-25: The Executive Interview; FAB: Frontal Assessment Battery; FOG-Q: Freezing of Gait Questionnaire; HY: Hoehn & Yahr; BD: Binswanger’s Disease; BI: Bartels Index; IPD: Idiopathic Parkinson’s Disease; MMSE: Mini Mental State Examination; MoCA: Montreal; Cognitive Assessment test; MRI: Magnetic Resonance Imaging; MUGA: Multigated Angiography; NDD: Nigrostriatal Dopaminergic Denervation; NINDS-AIREN: Association International pour la Recherché et l’Enseignement in Neurosciences and Hachinski Scores; PBMT: Intracerebral Transcatheter Laser Photobiomodulation Therapy; rTMS: repetitive Transcranial Magnetic Stimulation; REG: rheoencephalography; SG: Scintigraphy, reg: rheoencephalography; SPECT: Single Photon Emission Computed Tomography; SSS: Scandinavian Stroke Scale; UPDRS: Unified Parkinson’s Disease Rating Scales; UPSIT: Smell Identification Test; VP: Vascular Parkinsonism.

**Table 3 brainsci-13-00489-t003:** Studies results based on imaging techniques.

Study	Image Testing	Response to Treatment (Levodopa) Depending on Image	Comments
**Navarro-Otano, J. et a, 2014** [18]	123I-MIBG cardiac image	VP patients with normal H/M ratio (non-suggesting IPD): 0% (0/7) VP patients with low H/M ratio: 28.6% (2/8)	A normal H/M ratio (not suggestive of IPD) predicted a poor response to treatment.
**Lee, M.J. et al. 2015** [20]	[18F] FP-CIT PET	Group I: 4.5 (1/22) Group II: 40.0% (8/20)	Patients with a pathological PET study showed significantly better response to levodopa Good response based on ≥30% changes in UPDRS.
**Benítez-Rivero, S. et al., 2012** [15]	123 I-FP-CIT SPECT	Does not compare the response to treatment according to an image.	SPECT results were only associated with the presence of falls.
**Zijlmans, J. et al., 2007** [11]	[123I] FP-SPECT Based on BP%	Two L-dopa responders with a BG BP% similar to the 11 non-responders (mean 29.5 (28.4–30.5) vs mean 26.0 (6.9–56.5))	[123I] FP-SPECT uptake not correlated to levodopa response based on reduction in UPDRS III scale.
**Antonini, A. et al., 2012** [12]	FP-CIT SPECT	SPECT (no pathological) l: 93% (26/28) SPECT (pathological): 48% (23/48)	They confirm that a normal FP-CIT SPECT is associated with a poor levodopa effect.

(123I)-MIBG cardiac gamma-graphy: (123)I-metaiodobenzylguanidine on cardiac gammagraphy; (123I) FP-CIT SPECT: Single Photon Emission Tomography with 123Ioflupane; 18F-FDG-PET: Fluorodeoxyglucose labeled with 18F Positron Emission Tomography; BP: radiotracer pickup.

## Data Availability

The data presented in this study are available upon request from the corresponding author.

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
