# Peer review of "Treatment of Vascular Parkinsonism: A Systematic Review"

_brainsci, 2023, doi:10.3390/brainsci13030489_

Round 1

Reviewer 1 Report

In the article entitled "Treatment of vascular Parkinsonism: A Systematic Review" the authors did an interesting review of patients with vascular Parkinsonism who do not respond to L-Dopa treatment.

Some points to verify.

The article by Sato Y., 2013, was retracted, therefore it would be necessary to check if section 3.2 on vitamin D is correct or a modification would be required.

Also check if it will be taken into consideration in the systematic review.

Minor points.

Throughout the article authors abbreviate Vascular Parkinson's with VP, but there are sections where authors abbreviate it PV, please check.

Author Response

We would like to thank you for the information on the retraction of the article by Sato et al. After investigating the cause, we have found out that it was due to the inclusion among the authors of doctors who had not really participated in the study and were included as honorary authors.
The veracity of the data obtained and their analysis has not been questioned. We have included in the manuscript a small analysis of the risk of bias of the included articles.
For this reason we would like to maintain the data provided by this article, since the retraction is not due to the veracity of these and we consider that they add value to the  review.

We have reviewed all VP acronyms in the manuscript, again thank you for your review.

Reviewer 2 Report

The aim of the study "Treatment of vascular Parkinsonism: A Systematic Review" is to update and analyze the different treatments and their efficacy in vascular parkinsonism by literature search and a systematic review of 12 original studies. The results suggest that the response of patients to different therapeutic strategies is modest, with a subgroup of patients that can be identified as more responsive to L- dopa and which should be treated with appropriate doses.

The manuscript is well-structured and clear and concise.

There is an abbreviation (PV) throughout the entire manuscript that was not mentioned earlier. Probably it refers to VP.

Line 33: "The term vascular parkinsonism (VP) is one of the most controversial in neurology...". What is controversial about this term?

It's a little bit hard to follow tables, due to uneven titles and formatting. Also, Table captions (Study characteristics for example) should be improved since they should contain more details. 

Overall, there the Manuscript is relevant to the field and to the patients.

Author Response

Thank you for detecting the error in the abbreviation of VP, all of them have been revised.
The controversy described in the introduction about the term Vascular Parkinsonism has been explained in more detail.
The tables have been revised and simplified, and their titles have been edited.
The English of the manuscript has been checked.